# DafnyBench: A Benchmark for Formal Software Verification

**Chloe Loughridge**[*]                                                          *cloughridge@college.harvard.edu*
*Harvard College*

**Qinyi Sun**[*†]                                                                      *wendysun@mit.edu*
*Massachusetts Institute of Technology*

**Seth Ahrenbach**                                                      *seth.ahrenbach@omnifederal.com*
*                    *

**Federico Cassano**                                                      *cassano.f@northeastern.edu*
*Northeastern University*

**Chuyue Sun**                                                                *chuyues@stanford.edu*
*Stanford University*

**Ying Sheng**                                                                  *ying1123@stanford.edu*
*Stanford University*

**Anish Mudide**                                                                  *amudide@mit.edu*
*Massachusetts Institute of Technology*

**Md Rakib Hossain Misu**                                                            *mdrh@uci.edu*
*University of California Irvine*

**Nada Amin**                                                            *namin@seas.harvard.edu*
*Harvard University*

**Max Tegmark**                                                                  *tegmark@mit.edu*
*Massachusetts Institute of Technology*

**Reviewed on OpenReview:** *https://openreview.net/forum?id=yBgTVWccIx*

## Abstract

We introduce DafnyBench, the largest benchmark of its kind for training and evaluating machine learning systems for formal software verification. We test the ability of LLMs such as GPT-4 and Claude 3 to auto-generate enough annotations for the Dafny formal verification engine to successfully verify over 750 programs with about 53,000 lines of code. The best model and prompting scheme achieved 68% success rate, and we quantify how this rate improves when retrying with error message feedback and how it deteriorates with the amount of required code and

annotations. We hope that DafnyBench will enable rapid improvements from this baseline as LLMs and verification techniques grow in quality.

## 1 Introduction

Rapidly improving Large Language Models (LLMs) Bubeck et al. (2023); Anthropic (2024); Team et al. (2023) are helping accelerate software development through co-pilots and other program synthesis tools. But how can we ensure that LLM-generated code meets our specifications and reliably does precisely what it is supposed to do? Indeed, this remains a persistent problem even with human-written code: major code-testing efforts failed to prevent e.g. bugs causing an Ariane-V rocket explosion European Space Agency (1996) and embarrassing security vulnerabilities in ssh Heartbleed (2024) and the Bash shell Wikipedia contributors (2024b). The latter was built into the Unix operating system for 25 years before being discovered.

Although *formal verification* can guarantee perfect reliability, providing rigorous mathematical proof that software meets specification, it has yet to gain widespread adoption because it is costly. Formally verifying code is often a significant burden on the developer (Huang et al., 2024; Orenes-Vera et al., 2023). Moreover, existing formal-verification tools tend to involve a major learning curve above and beyond just learning to code, greatly reducing the pool of people able to do this work.

Machine learning methods have the potential to minimize a common pain point of formal methods, i.e., writing and verifying formal specifications. There is a growing body of work that demonstrates the effectiveness of LLMs on the analogous problem of automated theorem proving. In this related setting, AI produces formal proofs not about code but about mathematical theorems. Fueled by the advent of benchmarks totaling over 100,000 theorems, AI tools have during the past few years improved their proof success fraction to over 82% (Polu & Sutskever, 2020; Lample et al., 2022).

Unfortunately, formal verification sorely lacks correspondingly large benchmarks: the largest of their kind are *Clover* (Sun et al., 2024) and *dafny-synthesis* (Misu et al., 2024), containing 66 and 153 programs, respectively. There is room for expanding not only their size, but also their level of difficulty: For example, *Clover* is limited to single-function programs, and sometimes the formal specification for the program directly repeats the implementation of the algorithm (see Appendix G). To support automation of formal verification, the goal of the present paper is to provide such a benchmark expansion. We do so by assembling a suite of formally verified programs written in *Dafny*, a formal verification language that was developed for easy adoption by programmers due to its similarity with popular imperative programming languages such as Python and C++ (Leino, 2023). In order for formal verification to succeed, most of these programs require supplementary text constituting "annotations" to the automated theorem prover.

The rest of this paper is organized as follows. We summarize related work in Section 2, describe our benchmark construction in Section 3, and quantify the ability of current LLMs to solve benchmark verification tasks in Section 4. We summarize our results and discuss promising opportunities for further work in Section 5 . We provide further details on the benchmark construction and evaluation in appendices.

## 2  Related Work

The verification of software systems relies on a variety of approaches and frameworks. For proving full correctness, Hoare logic represents one of the most widespread formal frameworks (Huang et al., 2024). To prove the full correctness of a program using Hoare logic, one must give a specification for a program. Consider the program `LinearSearch(A, P)`, which finds the first element in the array A with some property P. The specification for this program consists of an optional precondition– which is a property of x in its original state, at the time of the function call– and a postcondition, which expresses a property of the program's result. In Figure 1, we define multiple postconditions for `LinearSearch` by using `ensures` statements on lines 2-4: (line 2) the returned index value from this function will be within the array bounds of A, (line 3) the value in A[n] has the desired property P and otherwise n is set to the length of A, (line 4) n is the first index at which P applies in the array A. We call this combination of precondition(s) and postcondition(s) the specification of the program. Verifying the program means proving that the implementation of the program matches its specification. Leino et al have designed a language for writing formally verifiable code, called Dafny (Leino, 2023), and this is the language in which we have written `LinearSearch` in Figure 1. In Dafny, a programmer declares specifications for methods using `ensures` and `requires` clauses. Programs in Dafny will only compile if an underlying SMT solver can find a proof showing that the program implementation matches its specification. Often the SMT solver requires additional annotations, like loop invariants and assert statements in the body of the program, to find a proof linking the program implementation to its specification.

```
method LinearSearch<T>(a: array<T>, P: T -> bool) returns (n: int)
    ensures 0 <= n <= a.Length
    ensures n == a.Length || P(a[n])
    ensures forall i :: 0 <= i < n ==> !P(a[i])
{
    n := 0;
    while n != a.Length
        invariant 0 <= n <= a.Length
        invariant forall i :: 0 <= i < n ==> !P(a[i])
    {
        if P(a[n]) {
            return;
        }
        n := n + 1;
    }
}
```

Figure 1: An example `ground_truth` program that is fully verified with Dafny. To create the `fill_verification_conditions` task, we would remove the `invariant` lines from the program above.

As summarized in Table 1 below, there is a striking lack of training data for formal verification: while there are hundreds of thousands of training examples for proving mathematical theorems and over ten thousand training examples for synthesizing programs, there is far less training data for formal verification — for example, there are only $66 + 153 = 219$ for proving program correctness in the Dafny language. This motivates our work in the current paper to expand

the benchmarks from *Clover* and *dafny-synthesis* and build DafnyBench. Mathematical theorem proving datasets focus on logical reasoning, but are disconnected from real-world programming applications. DafnyBench, as a formal verification benchmark, tests a model's capabilities in both formal reasoning and programming. Program synthesis benchmarks usually ask a model to generate code from specifications or descriptions of what the code is supposed to achieve, but they lack the correctness guarantee that verifies the generated code is correct. In contrast, DafnyBench asks a model to generate annotations from specifications and code, where "annotations" are supplementary text that can help Dafny verifier prove that a property claimed to be true is indeed true.

Table 1: Summary of popular machine-learning benchmark datasets for proving mathematical theorems, synthesizing programs, and formally verifying programs. Size is measured by the number of samples in each dataset. In the formal reasoning datasets, each sample is usually a math problem or a theorem. In the program synthesis and verified software programming benchmarks, each sample corresponds to a program.

| Category | Dataset | Size |
|---|---|---|
| **Mathematical theorem proving** | CoqGym (Yang & Deng, 2019) | 71,000 proofs |
| | LeanDojo (Yang et al., 2023) | 98,734 proofs |
| | PISA (pis, 2021) | 138,000 proofs |
| | Natural Proofs (Welleck et al., 2021) | 15,000 proofs |
| | Archive of Formal Proofs (Blanchette et al., 2015) | 1 million lines of code |
| **Unverified program synthesis** | APPS (Hendrycks et al., 2021) | 10,000 programs |
| | HumanEvalX (Zheng et al., 2023b; Chen et al., 2021) | 165 programs |
| | MBPP (Austin et al., 2021) | 974 programs |
| | SWEBench (Jimenez et al., 2023) | 2,294 programs |
| | LiveCodeBench (Jain et al., 2024) | grows weekly |
| **Formal software verification** | Clover (Sun et al., 2024) | 66 programs |
| | Dafny-synthesis (Misu et al., 2024) | 153 programs |

The 66 programs in the *Clover* benchmark are human-written. In contrast, *dafny-synthesis* translates 153 MBPP problems from Python to Dafny using GPT-4. While this method is more efficient than manual translation, it could potentially skew the distribution of represented problems away from real-world Dafny problems that may be too hard for GPT-4 to verify on its own (Misu et al., 2024). Our dataset counterbalances this potentially skewed distribution by introducing problems verified by human programmers on GitHub.

*Clover* proposes the most sophisticated benchmark evaluation strategy to date for formally verifiable software: the authors suggest a six-way consistency check between code, docstrings, and annotations. We do not yet implement the full *Clover* evaluation scheme in DafnyBench, and instead deem a benchmark program "solved" if a model can make it pass the Dafny verifier without modifying the `requires` and `ensures` statements in the program and without using `{:verify false}` or `assume false` (see Appendix F for further details).

## 3  DafnyBench Construction

### 3.1  Sourcing Ground Truth Programs

In total, our DafnyBench benchmark contains 782 `ground_truth` stand-alone Dafny programs that compile. These problems come from the following sources:

- **GitHub Scrape**: We scraped all publicly available Dafny files on GitHub published on the before the end of 2023. The relevant files were returned from the GitHub API using the `language:Dafny` search command. We adapted a deduplication script from (Mou et al., 2023) to retain a unique set of scraped Dafny files from Github. The de-duplication process reduced the number of `.dfy` files from ~15,000 to ~5,000. We then attempted to verify each of these remaining files using the `dafny verify` command with a local installation of Dafny 4.3.0, and removed any files that did not verify. At this stage, we removed all of the files from the *Clover* repository Sun et al. (2024), which had already been formatted as benchmark files. This left 1,112 files. We found that 374 of these files lacked *ensures* statements, and 459 of lacked `assert` and `invariant` clauses. We removed the union of these sets, which left us with 556 `ground_truth` files. Out of these files, 113 verify without any compiler annotations. To mitigate data contamination, models run on our benchmark should ideally not be trained on data from the repositories listed in Appendix E.

- **Clover**: We added 62 ground truth textbook Dafny programs provided by the *Clover* dataset (Sun et al., 2024). We formatted these to fit our benchmark style and removed their compiler annotations. Out of these files, 23 verify without any compiler annotations.

- **Dafny-synthesis**: Finally, we included 164 Dafny programs provided by the *dafny-synthesis* benchmark. These problems have been translated from the MBPP benchmark (Misu et al., 2024). Out of these files, 72 verify without any compiler annotations.

The `ground_truth` programs in our dataset have on average 2.04 methods, 0.98 functions, and 1.31 lemmas. This places the mean complexity of our examples at a level higher than *Clover* alone, which has only one stand-alone method per example.

### 3.2  Task Design: Fill Annotations

DafnyBench evaluates LLMs on the `fill_annotations` task. For this task, we took a `ground_truth` program, removed all of its annotations (i.e., all of the `assert` and `invariant` statements in the body of the code), and asked LLM to fill annotations back in so that the resulting program could be verified with Dafny.

We do not demarcate from where these annotations have been removed, i.e., we do not insert `/* TODO */` after we remove each annotation, which would make the task easier and not reflective of models utility in real-world use cases.

In the context of our running example of the `LinearSearch` program from Figure 1, completing the `fill_annotations` task would mean adding back the loop invariants– or some equivalent phrasing of them– on lines 8 and 9. Without these loop invariants, the program will not compile because Dafny cannot find a proof that it matches its specification.

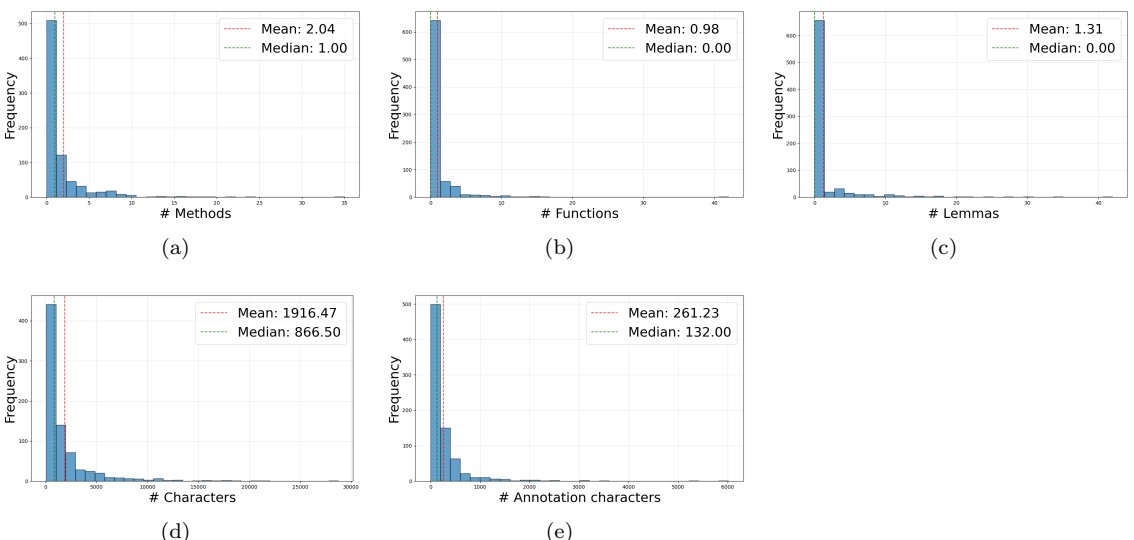

Figure 2: Distributions of method (a), function (b), lemma (c), character (d), annotation character (e) counts in DafnyBench.

**Evaluation Metric**   An LLM's attempt to fill annotations back in for a test program is counted as a success if all following conditions are satisfied: 1) The reconstructed program is verified with Dafny; 2) LLM preserves all preconditions (`requires` statements) and postconditions (`ensures` statements); and 3) LLM does not use `{:verify false}` or `{assume false}` to "cheat."

## 4   Experiments

In this section, we report success rates for different models on the `fill_annotations` task, as well as provide some insight into current LLMs' capabilities at writing annotations for formal verification.

### 4.1   Prompts & Hyperparameters

We tried to keep prompts and hyperparameters mostly the same across models in order to reduce the difference between model performances that is caused by hyperparameters. However, the prompts are not fully identical. For example, when we ask LLM to simply return the annotations-filled program without any explanation, Claude 3 tends to add explanations that interfere with Dafny compilation. Thus, we had to adjust some prompts slightly to fit each model's peculiarities.

For hyperparameters, we set `max_tokens` $= 4096$, which corresponds to the lowest max output token limit among all the evaluated models, and we set `temperature` $= 0.3$. We gave each model up to $n = 10$ attempts at a given file. If it succeeded on an attempt before the $n^{\text{th}}$, it would be early stopped. If the model failed on any of the intermediate attempts, it received the Dafny error message and was asked to fill in the annotations again with the error message taken into consideration. If it failed on all $n$ attempts, it was considered to fail on that specific test program.

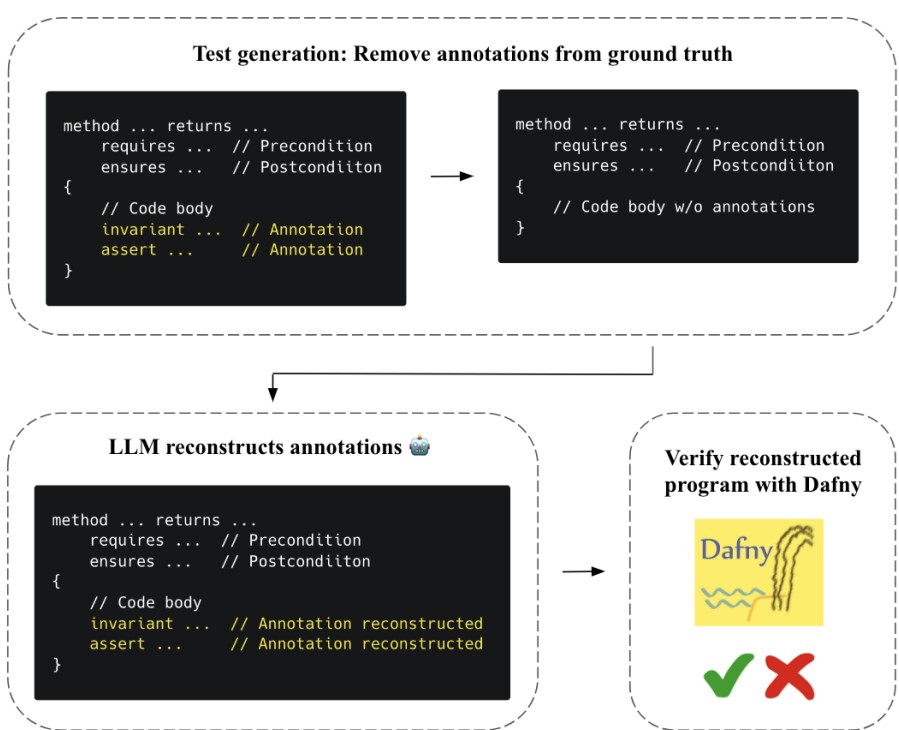

Figure 3: Overview of evaluating LLM on a DafnyBench test program.

## 4.2 Basic Results

We tested GPT-4o, GPT-4 Turbo OpenAI et al. (2024), GPT-3.5 Turbo Brown et al. (2020), Claude 3 Opus Anthropic (2024), and CodeLlama-7b-Instruct-hf hug (2022) on the 782-program benchmark. Table 2 shows that Claude 3 Opus performed best, achieving a success rate $\sim 68\%$.

## 4.3 Difficulty Utilizing Dafny Error Messages

Figure 4 shows how the cumulative success rate improved with more attempts $n$. We see that the best models succeeded on the first try about 54%, with rapidly diminishing returns after that, approaching a plateau about 65% for $n \sim 5$. This suggests that the LLMs are not great at taking Dafny error messages into consideration, or struggle to cope with the underlying task.

| Model | % Success |
|---|---|
| No LLM | 26.9 |
| GPT-3.5 Turbo | $44.0 \pm 1.8$ |
| GPT-4 Turbo | $59.8 \pm 1.8$ |
| GPT-4o | $59.3 \pm 1.8$ |
| Claude 3 Opus | $\mathbf{67.8} \pm 1.7$ |
| CodeLlama-7b-Instruct-hf | $28.0 \pm 1.6$ |

Table 2: Models' success rates at writing annotations for DafnyBench, with $n = 10$ attempts given. Dafny succeeds in auto-verifying some programs even without annotations, corresponding to the "No LLM" 26.9% success rate baseline.

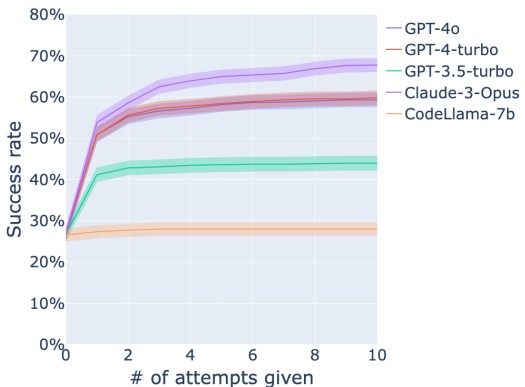

Figure 4: Success rate vs. number of attempts given.

### 4.4 Difficulty Grows with Program Size & Annotation Quantity

Figure 5a show that the success rate drops with program size. An obvious explanation could be that there is more to verify and more annotations needed. Also, as a program gets longer, there may be more dependencies among variables, functions, methods, and classes, increasing the overall verification difficulty level.

Figure 5b shows that the success rate drops with the annotation quantity, defined as the number of characters in the lines of compiler annotations. In other words, the success rate drops with the amount of work that the LLM needs to do (the amount of text that it needs to insert in the right places).

### 4.5 Models' Common Failure Types

To analyze where LLMs failed on the benchmark, we categorized failures into nine types, including verification logic error, code logic error, type error, resolution error, syntax issue, altered specification, timeout, trivial verification, and others. For a test program that a model failed at, we: 1) checked for timeout, cheating by altering specification, and cheating by trivial verification; and 2) passed Dafny error message from the failed program to Claude and asked it to classify the failure type. Table 3 explains each failure type, and Figure 6 gives by-model statistics of failure types.

## 5 Discussion & Conclusions

We have assembled the largest machine learning benchmark to date for formal software verification and made it publicly available on GitHub at `https://anonymous.4open.science/r/DafnyBench-839D`.

### 5.1 Opportunities for Larger Benchmarks

It will be valuable to further expand formal verification benchmarks, which still remain more than two orders of magnitude smaller than corresponding benchmarks for mathematical theorem proving.

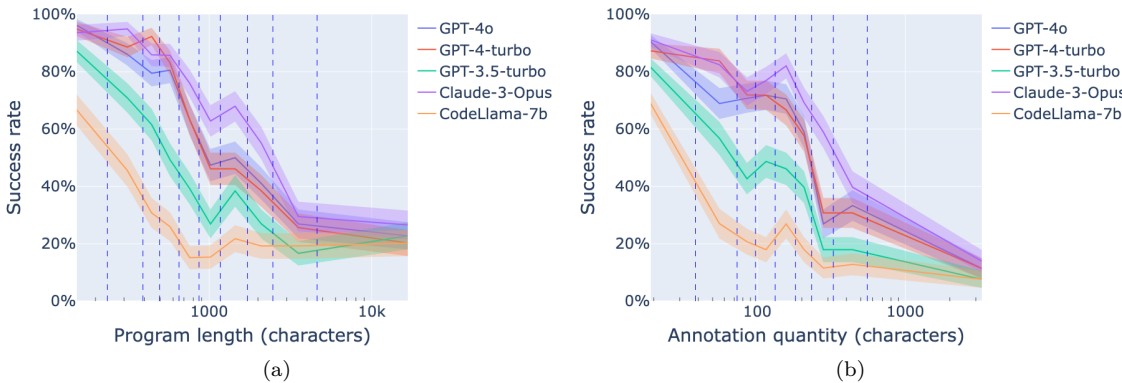

Figure 5: **Mean success rate of each bin vs. program length (a)**, and **mean success rate of each bin vs. annotation quantity (b)**. The vertical lines indicate the bin boundaries used, where the bins have an almost uniform distribution of the programs. Note that the bins are different for the two metrics. For better visual clarity, the scales are adjusted for both plots and their $x$-axes do not start at 0 character.

Table 3: **Examples of failure types**. Note that the examples are samples, not a complete list, for each failure type.

| Failure Type | Examples |
|---|---|
| Code logic error | Index out of range / Target object might be null |
| Verification logic error | Cannot prove termination / Assertion might not hold |
| Syntax issue | lbrace/rbrace expected / Semicolon expected / Unresolved identifier |
| Type error | Value does not satisfy the subset constraints of 'nat' |
| Resolution error | Boogie program had... resolution errors |
| Timeout | Verification timeout |
| Trivial verification | Cheating by using `{:verify false}` or `assume false` |
| Altered specification | Cheating by altering provided specification |
| Other | Failure type not belonging to any listed category above |

One convenient way to expand the number of available problems may involve incorporating Dafny programs from GitHub that have dependencies spread across multiple files (while DafnyBench encompasses increasingly complex multi-step programs, its programs each fit in a single file, avoiding the intricacies associated with distributed files or the integration of external libraries).

Perhaps models that perform especially well on this initial benchmark can later be used to expand it by translating existing Python benchmark problems into Dafny, Rust Klabnik & Carol Nichols (2021) or other popular formal verification languages.

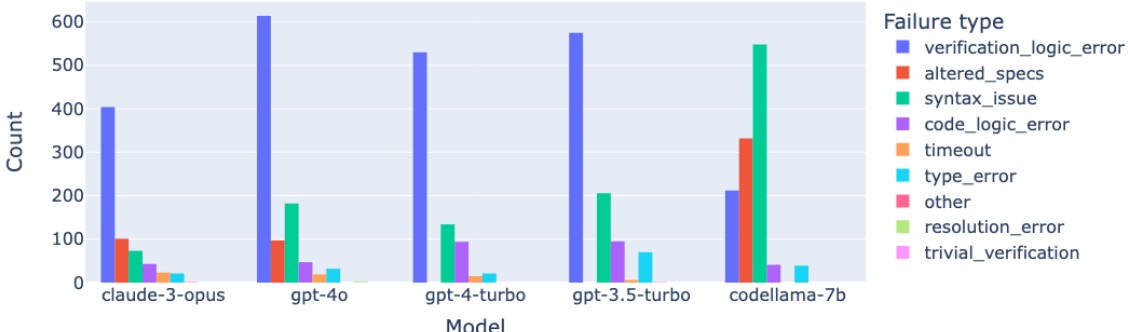

Figure 6: **Counts of failures by failure type and by model**. Note that a model could have multiple failures for a single test program (for example, it might have both verification logic error and syntax issue). Also note that the closed-source models had most of their failures at verification logic, while the open-source model had most of its failures at syntax issues and cheating by altering specification.

A subset of the programs we scraped from GitHub do not have appropriate docstrings. By building a benchmark with better code documentation, models may be able to leverage helpful contextual information to better constructing verification annotations.

## 5.2 Benchmark Evaluation Limitations

Data contamination emerges as a potentially significant limitation for evaluating LLMs on Dafny-Bench. Scraping data from platforms such as GitHub introduces risks of leveraging previous models' training data into the benchmark evaluation, potentially inflating the abilities of certain models. To help relieve this concern, future work can augment the programs in DafnyBench by doing program transformations (e.g., by changing variable names in the programs, adding spurious code blocks, and changing order of lines that do not depend on each other), or expand the dataset by asking LLMs that are good at Dafny or fine-tuned on datasets like DafnyBench to generate synthetic data.

Another limitation emerges in that DafnyBench does not assess a model's competence in translating natural language into concise formal specifications. Arguably, this conversion is a demanding and crucial skill we seek from language models: the capacity to validate, beyond merely verifying code. The pivotal question is whether a model can assist in identifying the essential properties an algorithm must fulfill. This provides an exciting frontier for future work, which we begin to brainstorm in Appendix B.

## 5.3 Opportunities for Improved LLM Results

We evaluated the models with a fixed temperature setting and a max output token limit of 4096, and we used prompts that were manually but not very systematically tuned for effectiveness (see Appendix A) — all of these choices probably leave room for improvement.

We do not yet provide an official training dataset or models custom-trained to do well on the DafnyBench evaluation set. However, we do provide the full json file produced by the GitHub scrape, and we separately provide the names of the files we use for the evaluation benchmark. Hence, it is possible for researchers to use files from the Github scrape that are not used in the benchmark as training data, though we cannot at this time provide strong guarantees on similarity between such training problems and the benchmark problems.

We also see opportunities for LLM-related innovation on the algorithmic side: out-of-the-box LLMs provide a floor but not a ceiling for possible performance on this benchmark. For example, fine-tuning or search-based inference-time algorithms might boost models' performances on this benchmark (Brandfonbrener et al., 2023).

### 5.4   The Potential of Better LLM-Powered Verifiers

LLMs also have potential to improve formal verification in more profound ways than mentioned above, when used in combination with other AI tools. For example, they can help automate the identification of sub-goals and annotations, reducing the search space for automated theorem provers and SAT solvers. A software developer is likely able to specify the high level assurance properties of a piece of code, but may lack familiarity with the complexities of proof sub-goals and annotations. LLMs offer a way to bridge this gap between software developers and formal verification.

Bigger, more general benchmarks can be used to train LLMs to specify sub-goals and annotations in formats most useful to the presently available provers and solvers. Benchmarks covering broad ground, from cryptography, lambda calculus, embedded systems, and avionics, in a variety of widely used programming languages suitable for verification, will help create LLMs that can take real-world software, automatically process and serve it to verification tools, and inform the developer in near real time about the correctness of the code. The problem is analogous to that solved by existing automated theorem provers and model checkers in the domain of mathematics. For a survey on the application of deep learning to automated theorem proving, see Li et al. (2024).

For further discussion on LLM's potential for auto-verifying program synthesis, see Appendix C.

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

## A   Prompt Engineering for Annotation Reconstruction

We based our prompts on the prompts used in the *Clover* benchmark (Sun et al., 2024), one of the previously largest such benchmarks, since they provide a fairly rigorous precedent. We tried to keep prompts mostly the same across models in order to reduce the difference between model performances that is caused by prompts. However, the prompts are not fully identical. For example, when we ask LLM to simply return the annotations-filled program without any explanation, Claude 3 tends to add explanations that interfere with Dafny compilation. Thus, we had to adjust some prompts slightly to fit each model's peculiarities.

### A.1   GPT Model Family Prompts

```
SYSTEM_PROMPT = ''You are an expert in Dafny. You will be given tasks dealing
                  with Dafny programs including precise annotations.''

USER_PROMPT = ''Given a Dafny program with function signature, preconditions,
                postconditions, and code, but with annotations missing.
                Please return a complete Dafny program with the strongest
                possible annotations (loop invariants, assert statements,
                etc.) filled back in. Do not explain. Please use exactly the
                same function signature, preconditions, and postconditions.
                Do not ever modify the given lines. Below is the program:''
```

### A.2   Claude 3 Opus Prompts

```
SYSTEM_PROMPT = ''You are an expert in Dafny. You will be given tasks dealing
                  with Dafny programs including precise annotations. You should
                  only return code body in all circumstances. No text is allowed.''
```

```
USER_PROMPT = ``Given a Dafny program with function signature, preconditions,
              postconditions, and code, but with annotations missing.
              Please return a complete Dafny program with the strongest
              possible annotation (loop invariants, assert statements,
              etc.) filled back in. Do not explain or output any text. If
              you have to explain, put all explanations in comments form.
              There should only be code body in your output. Please use
              exactly the same function signature, preconditions, and
              postconditions. Do not ever modify the given lines. Below
              is the program:\n```dafny\n''
```

### A.3 CodeLlama-7b-Instruct-hf Prompts

The prompts for CodeLlama-7b-Instruct-hf are the same as those in A.2.

## B  Proposals for Evaluating Strength of Generated Specifications

The evaluation of models' capability to generate formal specifications might be enhanced by integrating the process with the creation of positive and negative test cases for each Dafny implementation. This approach proposes a reward system where models are evaluated based on the number of positive test cases their formal specifications support and the number of negative test cases they successfully reject. However, this method introduces a new challenge: ensuring the test cases accurately reflect the comprehensive meaning intended in the natural language descriptions. The consistency and validity of these test cases become critical, raising questions about the methods used to generate and verify them.

## C  Further Discussion

### C.1  The Potential of Auto-Verifying Program Synthesis

Above we discussed the challenge of verifying existing pre-programs. Another potential of LLMs is to use program-synthesis techniques that produce both programs and proofs of their correctness, all at the same time. This makes intuitive sense, since when a human programmer writes code, they typically have an informal proof in their head for why this code is correct. In other words, in addition to bridging the gap from low level implementation to high level specification in the upward direction, LLMs can offer assistance in generating provably correct low level code from high level specifications via program synthesis.

Current approaches to program synthesis enable engineers to encode a desired specification in a high level language, and then through a (hopefully) verified correct compiler generate correct low level code in a language like VHDL Committee & Subcommittee (2019) or Verilog Thomas & Moorby (2008) for hardware synthesis. Program synthesis is limited by the need for a special purpose language or compiler to be constructed and verified correct in its own right. For example, ReWire, a domain specific language defined as a subset of Haskell Procter et al. (2015), was manually verified correct using the Coq Interactive Theorem Prover. In order to add a new high-to-low path, a new

language or compiler will need to be defined and verified. If an engineer needs to synthesize correct Verilog rather than VHDL, they would likely need to first learn Caisson Li et al. (2011).

LLMs offer a way to generalize this approach. Starting with a high level language, an engineer might be able to specify a system and then leverage a LLM to generate low level code with the corresponding loop invariants, weakest pre-conditions, strongest post-conditions, etc, included. Early results indicate that an LLM that is able to converse with a human when producing a program can reduce the error rate against a simple programming benchmark by half Austin et al. (2021). If instead of receiving feedback from a human, the LLM were to interact with a suite of formal verification tools, we expect further improvements. The LLM should be capable of generating code that is appropriately annotated for theorem proving, which is exactly the skill assessed by test benches like that described here.

## D    The Minhash Deduplication Algorithm

We can think about deduplicating a set of files by finding groups of "similar"files and then choosing only one file representative from each group to form our final deduplicated set of files. To do this, we can use the Jaccard similarity metric to decide whether one document is a duplicate of another.

The Jaccard similarity metric provides a way to quantify the similarity of two sets. It is defined as (Wikipedia contributors, 2024c):

$$J(A, B) \,=\, \frac{|A \cap B|}{|A \cup B|}$$

In the application to code files, we could consider each file to be a set of $n$-grams, where an $n$-gram is defined as a sequence of $n$ adjacent symbols in a particular order (Wikipedia contributors, 2024a), and then apply the Jaccard score as a similarity metric for our files. To directly calculate this Jaccard score, we would need to run string comparison on every $n$-gram, which would have time complexity $O\left(nm^2\right)$ if we have $n$ $n$-grams each with max length $m$ characters. This turns out to be an inefficient method for representing each code file as a set. Instead, the minhash deduplication algorithm approximates the Jaccard similarity between two documents by shingling the documents and comparing the minhash representation of each set of shingles (i.e. we compare fingerprints of documents instead of full documents). The minhash representation of a document is a way to represent a text document as a set of numbers that is faithful to the structure of its content but with a fixed set size that is smaller than the total number of $n$-grams in the document (i.e. the minhash representation of the document is a form of numerical fingerprint of the document). In Figure 7 below, we provide the pseudocode for the minhash algorithm used, based entirely on the script in (Mou et al., 2023):

Note that the probability two files have the same min hash value under the same hash function is equivalent to their Jaccard similarity. Concretely, for file $A$ and file $B$:

$$\Pr\left[\min h_i(A) = \min h_i(B)\right] \,=\, J(A, B)$$

where $\min h_i()$ denotes taking the minimum hash value under hash function $h_i$. This makes sense because, assuming negligible hash collision, $\Pr\left[\min h_i(A) = \min h_i(B)\right]$ is equivalent to the

```
function minhash_deduplication(documents, num_permutations, threshold):
    # Preprocess the documents
    for each document in documents:
        tokenize the document into n-grams (shingles)
        hash each n-gram using a hash function (e.g., xxHash or SHA-1)
        store the hashed n-grams in a set

    # Generate permutations
    for i from 1 to num_permutations:
        generate random coefficients a and b
        create a permutation function: (a * x + b) % prime_modulus

    # Create minhash signatures
    signatures = []
    for each document in documents:
        signature = []
        for each permutation function:
            min_hash = INFINITY
            for each hashed n-gram in the document:
                permuted_hash = apply permutation function to hashed n-gram
                min_hash = min(min_hash, permuted_hash)
            append min_hash to signature
        append signature to signatures

    # Perform Locality-Sensitive Hashing (LSH)
    # We use 250 permutations, so to achieve Jaccard similarity threshold of 0.5
    # We really only need one band (i.e. one hash table)
    num_bands = choose number of bands
    rows_per_band = num_permutations / num_bands
    candidate_pairs = []
    for each band:
        create an empty hash table
        for each document signature:
            band_signature = subset of signature for the current band
            hash_bucket = hash(band_signature)
            add document to the corresponding hash bucket
        for each hash bucket:
            if number of documents in the bucket > 1:
                generate all pairs of documents in the bucket
                add pairs to candidate_pairs

    # Use a union-find datastructure to track groups of duplicates
    duplicates = UnionFind()
    for each band:
        for each row in hashtable:
            for each hash_bucket:
                if size(hash_bucket) <= 1:
                    continue
                else:
                    cluster_id = min(hash_bucket)
                    for x in hash_bucket:
                        duplicates.union(x, cluster_id)

    # Perform deduplication
    deduplicated_documents = []
    for each document in documents:
        if duplicates.find_root(document) = document:
            add document to deduplicated_documents

    return deduplicated_documents
```

Figure 7: Pseudocode for the minhash deduplication algorithm (continued).

probability that the first $n$-gram hash of $A$ under $h_i$ is equal to the first $n$-gram hash of $B$ under $h_i$. If $h_i$ is a good hash function, then it uniformly distributes the hash values of the original n-gram hashes over the range of $h_i$. Let $c$ denote the number of $n$-grams with equivalent hashes; let $a$ denote the number of $n$-grams from $A$ with smaller hash values than the hash value of corresponding $n$-gram from $B$; let $b$ denote the reverse of the previous category. Then, $\Pr\left[\min h_i(A) = \min h_i(B)\right] = \frac{c}{a+b+c}$, given the uniformity of $h_1$. Note that $\frac{c}{a+b+c} = \frac{|A \cap B|}{|A \cup B|} = J(A, B)$.

## E   Repositories of Scraped Dafny Code

We provide a full list of all repositories whose data we used in the scraped portion of DafnyBench in Tables 4, 5, 6. When reporting the license information, "Renamed so N/A" implies that the original repository we scraped in December 2023 no longer exists under that name. Otherwise, the repositories have either Microsoft open-source licenses, MIT licenses, GNU General Public License v3.0 licenses, Creative Commons Zero v1.0 Universal, Apache 2.0 licenses, or "Other" (which is secretly an MIT License in a strange format, which has been checked manually). In light of this, we release our derivative DafnyBench repository under an Apache 2.0 license and a GNU General Public License v3.0. We note explicitly here that all files from repositories with the Apache 2.0 license have been modified from their original form.

## F   Dafny Verification Examples

We take one example test program from DafnyBench, and consider four possible results for the corresponding LLM-reconstructed program: successfully verifies, fails to verify, cheats by including `assume false`, and cheats by including `{:verify false}`. The last three cases are all considered a fail by the DafnyBench evaluation metric.

### F.1   Successful Example

Figure 8 shows a Dafny program that is considered to have successfully verified without cheating.

**Dafny verifier message**: Dafny program verifier finished with 3 verified, 0 errors.

### F.2   Failed Example

Figure 9 shows a Dafny program that fails to be verified.

**Dafny verifier message**: (20,11): Error: index out of range. (30,4): Error: a postcondition could not be proved on this return path. (11,28): Related location: this is the postcondition that could not be proved. Dafny program verifier finished with 2 verified, 2 errors.

### F.3   Cheat Example

Figure 10 shows that a Dafny program cheats by including `assume false`, which DafnyBench evaluation would count as a fail.

**Dafny verifier message**: Dafny program verifier finished with 3 verified, 0 errors.

Table 4: Repositories from which DafnyBench utilizes scraped code (no particular order).

| Repository Name | License |
|---|---|
| dafl | No license provided |
| Dafny-Grind75 | No license provided |
| feup-mfes | MIT License |
| Dafny | GNU General Public License v3.0 |
| nitwit | MIT License |
| Dafny-experiences | No license provided |
| Formal_Verification_With_Dafny | No license provided |
| SENG2011 | No license provided |
| M2 | No license provided |
| assertive-programming-assignment-1 | No license provided |
| t1_MF | No license provided |
| dafny-exercise | Other |
| dafny-learn | No license provided |
| software-specification-p1 | No license provided |
| FMSE-2022-2023 | The Unlicense |
| fv2020-tms | No license provided |
| type-definition | No license provided |
| laboratory | No license provided |
| dafny | GNU General Public License v3.0 |
| TFG | GNU General Public License v3.0 |
| SiLemma | MIT License |
| dafny-training | No license provided |
| FormalMethods | No license provided |
| dafny_misc | MIT License |
| vmware-verification-2023 | No license provided |
| CSU55004—Formal-Verification | No license provided |
| MIEIC_mfes | MIT License |
| Dafny-programs | No license provided |
| MFES_2021 | MIT License |
| DafnyPrograms | No license provided |
| cs357 | No license provided |
| formal-methods-in-software-engineering | No license provided |
| Dafny_ProgrammingLanguages | No license provided |
| CSC8204-Dafny | No license provided |
| BPTree-verif | No license provided |
| tangent-finder | No license provided |
| Trab1-Metodos-Formais | No license provided |
| verified-using-dafny | MIT License |
| Metodos_Formais | No license provided |
| lets-prove-blocking-queue | Creative Commons Zero v1.0 Universal |
| Dafny_Programs | No license provided |
| dafny-workout | MIT License |

Table 5: Repositories from which DafnyBench utilizes scraped code (no particular order), continued.

| Repository Name | License |
| --- | --- |
| Dafny-Projects | No license provided |
| VerifiedMergeSortDafny | No license provided |
| dafny_projects | No license provided |
| pucrs-metodos-formais-t1 | No license provided |
| specTesting | No license provided |
| QS_BoilerPlate1 | No license provided |
| dafny-sandbox | No license provided |
| Formal-Verification | No license provided |
| dafny-duck | No license provided |
| FlexWeek | No license provided |
| 703FinalProject | No license provided |
| MFS | No license provided |
| dafny-mini-project | No license provided |
| Software-Verification | No license provided |
| circular-queue-implemetation | No license provided |
| Final-Project-Dafny | No license provided |
| DafnyProjects | No license provided |
| bbfny | No license provided |
| Formal-methods-of-software-development | No license provided |
| Software-building-and-verification-Projects | No license provided |
| software_analysis | No license provided |
| cs245-verification | No license provided |
| dafny-aoc-2019 | No license provided |
| ProjectosCVS | No license provided |
| MFDS | MIT License |
| groupTheory | No license provided |
| dafny-language-server | Other |
| Invoker | Apache License 2.0 |
| formal-verification | No license provided |
| dafny-programs | No license provided |
| ironsync-osdi2023 | Other |
| verified-isort | No license provided |
| paxos_proof | No license provided |
| se2011 | No license provided |
| Dafny_Verify | No license provided |
| Formal-Methods-Project | No license provided |
| 630-dafny | No license provided |
| dafny_examples | MIT License |
| Workshop | No license provided |
| Dafny-Practice | MIT License |
| CVS-handout1 | No license provided |
| CS494-final-project | No license provided |

Table 6: Repositories from which DafnyBench utilizes scraped code (no particular order), continued.

| Repository Name | License |
| --- | --- |
| iron-sync | Other |
| stunning-palm-tree | Creative Commons Zero v1.0 Universal |
| sat_dfy | No license provided |
| verification-class | MIT License |
| AssertivePrograming | No license provided |
| Dafny-VMC | MIT License |
| libraries | Other |
| cmsc433 | No license provided |
| Correctness | No license provided |
| CVS-Projto1 | No license provided |
| dafleet | MIT License |
| dafny-rope | MIT License |
| protocol-verification-fa2023 | No license provided |
| vfag | No license provided |
| Dafny_Learning_Experience | Apache License 2.0 |
| summer-school-2020 | No license provided |
| BinarySearchTree | Renamed so N/A |
| llm-verified-eval | MIT License |
| Programmverifikation-und-synthese | Renamed so N/A |
| Prog-Fun-Solutions | Renamed so N/A |
| CO3408-Advanced-Software-Modelling-Assignment... | Renamed so N/A |
| DafnyExercises | No license provided |
| test-generation-examples | No license provided |
| HATRA-2022-Paper | No license provided |
| veri-sparse | No license provided |
| Formal-Verification-Project | No license provided |
| formal_verication_dafny | No license provided |
| Simulink-To_dafny | No license provided |
| dafny_experiments | No license provided |
| cs686 | No license provided |
| Program-Verification-Dataset | MIT License |
| Dafny-demo | No license provided |
| dafny-exercises | No license provided |
| metodosFormais | No license provided |
| CS5232_Project | No license provided |
| Dafny-Exercises | No license provided |

```
function sorted(a: array<int>) : bool
    reads a
{
    forall i,j : int :: 0 <= i < j < a.Length ==> a[i] <= a[j]
}

method BinarySearch(a: array<int>, x: int) returns (index: int)
    requires sorted(a)
    ensures 0 <= index < a.Length ==> a[index] == x
    ensures index == -1 ==> forall i : int :: 0 <= i < a.Length ==> a[i] != x
{
    var low := 0;
    var high := a.Length - 1;
    var mid := 0;

    while (low <= high)
        invariant 0 <= low <= high + 1 <= a.Length
        invariant x !in a[..low] && x !in a[high + 1..]
    {
        mid := (high + low) / 2;
        if a[mid] < x {
            low := mid + 1;
        }
        else if a[mid] > x {
            high := mid - 1;
        }
        else {
            return mid;
        }
    }
    return -1;
}
```

Figure 8: An example response that successfully fills annotations back in and verifies without cheating.

### F.4 Another Cheat Example

Figure 11 shows that another Dafny program cheats by including `{:verify false}`, which Dafny-Bench evaluation would count as a fail.

**Dafny verifier message**: Dafny program verifier finished with 3 verified, 0 errors.

## G   Overdetailed Specification

Figures 12 and 13 show two example programs `update_array_strong.dfy` and `triple_strong.dfy` from the *Clover* benchmark (Sun et al., 2024), in which the formal specification closely echoes the program implementation.

```
function sorted(a: array<int>) : bool
    reads a
{
   forall i,j : int :: 0 <= i < j < a.Length ==> a[i] <= a[j]
}

method BinarySearch(a: array<int>, x: int) returns (index: int)
    requires sorted(a)
    ensures 0 <= index < a.Length ==> a[index] == x
    ensures index == -1 ==> forall i : int :: 0 <= i < a.Length ==> a[i] != x
{
    var low := 0;
    var high := a.Length - 1;
    var mid := 0;

    while (low <= high)
    {
        mid := (high + low) / 2;
        if a[mid] < x {
            low := mid + 1;
        }
        else if a[mid] > x {
            high := mid - 1;
        }
        else {
            return mid;
        }
    }
    return -1;
}
```

Figure 9: An example response that fails to be verified, as it lacks necessary `invariant` statements.

## H   Ethics Statement

In creating DafnyBench, we took care to use only data that was publicly available on GitHub, and we reference every repository from which we acquired this data, along with their licenses, in Appendix E. Furthermore, we cite the existing verifiable programming benchmarks that we subsume in DafnyBench (i.e. *Clover* Sun et al. (2024) and *dafny-synthesis* Misu et al. (2024)), and we asked explicit permission from their authors in order to do so. Finally, we cite all models that were used for evaluations on this benchmark OpenAI et al. (2024); Brown et al. (2020); Anthropic (2024); hug (2022). We used these models in accordance with the policies set forth in their API and model card documentation.

## I   Reproducibility Statement

Our benchmark contains the 782 `ground_truth` programs and the corresponding `verification_conditions_removed` programs. Additionally, we include full metadata on all of these files and the evaluation scripts necessary for running the listed models on them. By using the OpenAI and Anthropic APIs, others looking to reproduce this work should not expect to

```
function sorted(a: array<int>) : bool
    reads a
{
   forall i,j : int :: 0 <= i < j < a.Length ==> a[i] <= a[j]
}

method BinarySearch(a: array<int>, x: int) returns (index: int)
    requires sorted(a)
    ensures 0 <= index < a.Length ==> a[index] == x
    ensures index == -1 ==> forall i : int :: 0 <= i < a.Length ==> a[i] != x
{
    assume false;
    var low := 0;
    var high := a.Length - 1;
    var mid := 0;

    while (low <= high)
    {
        mid := (high + low) / 2;
        if a[mid] < x {
            low := mid + 1;
        }
        else if a[mid] > x {
            high := mid - 1;
        }
        else {
            return mid;
        }
    }
    return -1;
}
```

Figure 10: An example response that cheats by including `assume false`.

spend more than $300 for a full run of GPT-4o on DafnyBench, $300 for a full run of Claude3 on DafnyBench, $500 for a full run of GPT-4-turbo on DafnyBench, and $400 for a full run of GPT-3.5 on DafnyBench. We used the `sglang` package Zheng et al. (2023a) to efficiently query the models. All evaluations were completed on a Linux cluster with an A100 Nvidia GPU.

```
function sorted(a: array<int>) : bool
    reads a
{
   forall i,j : int :: 0 <= i < j < a.Length ==> a[i] <= a[j]
}

method {:verify false} BinarySearch(a: array<int>, x: int) returns (index: int)
    requires sorted(a)
    ensures 0 <= index < a.Length ==> a[index] == x
    ensures index == -1 ==> forall i : int :: 0 <= i < a.Length ==> a[i] != x
{
    var low := 0;
    var high := a.Length - 1;
    var mid := 0;

    while (low <= high)
    {
        mid := (high + low) / 2;
        if a[mid] < x {
            low := mid + 1;
        }
        else if a[mid] > x {
            high := mid - 1;
        }
        else {
            return mid;
        }
    }
    return -1;
}
```

Figure 11: An example response that cheats by including {:verify false}.

```
method UpdateElements(a: array<int>)
  requires a.Length >= 8
  modifies a
  ensures old(a[4]) +3 == a[4]
  ensures a[7]==516
  ensures forall i::0 <= i<a.Length ==> i != 7 && i != 4 ==> a[i] == old(a[i])
{
  a[4] := a[4] + 3;
  a[7] := 516;
}
```

Figure 12: An example program update_array_strong.dfy from the *Clover* benchmark (Sun et al., 2024), in which the formal specification closely echoes the program implementation.

```
method Triple (x:int) returns (r:int)
   ensures r==3*x
{
  r:= x*3;
}
```

Figure 13: Another example program `triple_strong.dfy` from the *Clover* benchmark (Sun et al., 2024), in which the formal specification closely echoes the program implementation.

