# OpenReview forum: "DafnyBench: A Benchmark for Formal Software Verification"
_TMLR — Accepted by TMLR_

### Review · Reviewer_d3dm · 2024-10-31

**Summary Of Contributions:**

This paper introduces DafnyBench, a dataset of programs written in Dafny, a formal
verification-focused programming language, along with an evaluation task which
measures a system's ability to infer annotations necessary for the compiler to
verify the given program.  The dataset comprises approximately 700 curated
programs collected from GitHub, a $6\times$ increased compared to a comparable
formal verification-focused dataset, yet far smaller than datasets on the
related but distinct task of formal mathematical theorem proofs.  The
evaluation metric that accompanies the dataset is a "complete the program" task
where the model (usually an LLM) being evaluated must provide annotations to
a program (e.g., assertions, loop invariants) such that the Dafny compiler can
verify the program constraints.  Finally, the paper tests the metric on
a handful of LLMs to give a sense of baseline performance on the task and what
insights the dataset and metric can produce.

**Audience:**

Yes

**Broader Impact Concerns:**

Table 5 shows that a large proportion of the code included in DafnyBench is
unlicensed.  While the strictest legal interpretations (at lest in places like
the United States) would suggest that one should not use unlicensed code, using
such code (especially when it is published on an open source-centric platform
like GitHub) is standard operating procedure for machine learning research (for
better or for worse).  That being said, I think it is important to note this
fact in the main body of the paper.  Furthermore, it is also important to note
if there is any material included that has licenses which notably differ from
free software license (e.g., non-commercial use licenses).

**Claims And Evidence:**

Yes

**Requested Changes:**

### Framing
The most important changes I would like to see in the paper are completing the
framing which is already present.  This starts with giving a more robust sense
of why the dataset is being introduced and how the proposed task/evaluation is
important to formal verification as field.  While the introduction does a good
job giving the context of formal verification generally, there is not as
concrete a sense given as to what DafnyBench will be used for.  Here are some
particular ways where answering this question is relevant:
- How is program synthesis generally (i.e., no formal verification involved)
  related or not related to what DafnyBench is addressing?  Clearly, there is
  much more data for plain program synthesis, so going through all the trouble
  of collecting Dafny data specifically should be justified by emphasizing the
  contrast between what we might use the Dafny data for vis-à-vis the
  limitations of what we can do with plain program synthesis data.
- This exact same question is touched on but not robustly addressed with
  respect to mathematical proofs: how are they the same? How are they
  different?  I am not saying that I actually think there is no difference but
  that by properly motivating and contrasting DafnyBench, it will help readers
  understand if DafnyBench is relevant for their own research and how it might
  be used down the road for research, say, unifying formal verification and
  computer-verifiable mathematical proofs.
- Is the evaluation task for DafnyBench only intended to be used with LLMs?  It
  seems like the annotation task _could_ be performed by a simpler model, but
  perhaps there are reasons for preferring the LLM (e.g., less task-specific
  engineering).  Again, I think making it more clear why DafnyBench is being
  introduced and what it will be used for will, in turn, make it clear why the
  task is and ought to be LLM-focused.


### Minor changes
- (p. 2) "Leino (2023)" -> "\citep"
- (p. 3) The paper mentions that the way _dafny-synthesis_ was created could give a skewed distribution, but then later DafnyBench incorporates the data anyway; it seems a bit incongruous.
- (p. 3) The description of how _Clover_ evaluates programs is a bit unclear.
  I would recommend either going into less detail and not attempting to
  describe its evaluation process or spend a little more time sketching it out,
  because I was unsure of what _Clover_ was doing after reading the paragraph
  (e.g., "Why would it reject valid programs?").
- (p. 5) The introduction to 3.2 is a bit abrupt: are we now talking about the
  task that DafnyBench uses to score the models?  A sentence or two introduce
  this section would ameliorate this.
- (p. 5) `"cheat"` $\rightarrow$ ``` ``cheat''```
- Sections 4.4 and 4.5 are a little sparse.  I think these sections should have
  a bit more content---maybe a full paragraph.  The explication does not have
  to be extensive, but there should be at least some indication of why these
  observations are interesting/important.

**Strengths And Weaknesses:**

### Strengths
This paper is a solid paper overall, differentiating itself from prior work
while also demonstrating appropriate continuity.  (This being said, I am not
familiar with the literature at the intersection of formal verification and
machine learning, so I might be unaware of relevant work.)  The nature and
structure of the resources being introduced is clear, well-defined, and easy to
follow; that is, I feel like I have a good sense of what DafnyBench is having
read the paper and how it could fill certain gaps in formal
verification-focused program synthesis.  The data collection and filtering is
well-documented and appropriate for resource being created.  The task is
clearly defined and is well-suited to the shape of the problem (e.g., using the
compiler to formally verify the program).  Finally the baseline methods and
accompanying analyses give a brief but informative sense of how difficult the
proposed task is and what next steps might be in improving performance on it.

### Weaknesses
I do not think there are any major weaknesses with the paper.  I believe the
most relevant weakness has to do with framing DafnyBench in terms of its
intended uses, making it more difficult to determine the standard by which to
judge the utility of the resource.  This is discussed more in the "Requested
Changes" section.

---

> ### Author Response · Authors · 2024-11-18
> **Response to Reviewer d3dm**
>
> Thank you for the detailed and constructive feedback! We are glad that you think this is a solid paper, and we appreciate your raising the concern about the overall framing. We reply to each of your suggestions in order.
>
> > How is program synthesis generally (i.e., no formal verification involved) related or not related to what DafnyBench is addressing?
>
> We add the following to Section 2 to make the context more clear. “Program synthesis benchmarks usually ask a model to generate code from specifications or descriptions of what the code is supposed to achieve, but they lack the correctness guarantee that verifies the generated code is correct. In contrast, DafnyBench asks a model to generate annotations from specifications and code, where 'annotations' are supplementary text that can help Dafny verifier prove that a property claimed to be true is indeed true." Unlike program synthesis benchmarks, DafnyBench tests not a model’s capability to simply write code, but a model’s capability to write annotations that help formally verify the code.
>
> > This exact same question is touched on but not robustly addressed with respect to mathematical proofs: how are they the same? How are they different?
>
> We also add the following to Section 2. “Mathematical theorem proving datasets focus on logical reasoning, but are disconnected from real-world programming applications. DafnyBench, as a formal verification benchmark, tests a model’s capabilities in both formal reasoning and programming.”
>
> > Is the evaluation task for DafnyBench only intended to be used with LLMs? It seems like the annotation task could be performed by a simpler model, but perhaps there are reasons for preferring the LLM (e.g., less task-specific engineering).
>
> DafnyBench is mainly intended to be evaluating LLMs. LLMs are widely used for program synthesis nowadays (e.g., Copilot, Cursor), but without formal verification, we can’t be sure the LLM-generated code does what it’s supposed to do. Thus, we hope to bridge this gap by having DafnyBench that evaluates a LLM’s capability at program verification. As shown in Table 3 on pg. 7, among the models we evaluated, only the frontier LLMs (Claude 3 Opus, GPT-4 Turbo, GPT-4o) were able to achieve success rate > 50%, and the smaller model CodeLlama-7b-Instruct-hf (a model specifically for generating code) only achieved a success rate that is ~1% above the 26.9% success rate baseline. Therefore, we don’t think non-LLM smaller models can consistently solve this task.
>
> > (p. 2) "Leino (2023)" -> "\citep"
>
> Fixed!
>
> > (p. 3) The paper mentions that the way dafny-synthesis was created could give a skewed distribution, but then later DafnyBench incorporates the data anyway; it seems a bit incongruous.
>
> _Dafny-Synthesis_ mostly contains programs that have 1 - 2 short methods, thus the overall distribution might be leaning towards easier programs. While DafnyBench incorporates programs from _Dafny-Synthesis_, the majority of the DafnyBench programs comes from GitHub scrape, which is on average longer (see Table 2) and closer to real-world Dafny problems. We also consider replacing Table 2 with histograms for methods / functions / lemmas / characters / annotation characters to better reflect the distributions of these quantities in DafnyBench.
>
> > (p. 3) The description of how Clover evaluates programs is a bit unclear. I would recommend either going into less detail and not attempting to describe its evaluation process or spend a little more time sketching it out, because I was unsure of what Clover was doing after reading the paragraph (e.g., "Why would it reject valid programs?").
>
> Great point! Introducing too much detail here about the _Clover_ evaluation process might confuse readers. We removed the part that describes the evaluation details, and only keep the following: “_Clover_ proposes the most sophisticated benchmark evaluation strategy to date for formally verifiable software: the authors suggest a six-way consistency check between code, docstrings, and annotations. We do not yet implement the full _Clover_ evaluation scheme in DafnyBench…”
>
> > (p. 5) The introduction to 3.2 is a bit abrupt: are we now talking about the task that DafnyBench uses to score the models? A sentence or two introduce this section would ameliorate this.
>
> We changed the beginning of Section 3.2 to “DafnyBench evaluates LLMs on the fill_annotations task.”
>
> > (p. 5) "cheat" -> ``cheat''
>
> Fixed!
>
> > Sections 4.4 and 4.5 are a little sparse. I think these sections should have a bit more content---maybe a full paragraph. The explication does not have to be extensive, but there should be at least some indication of why these observations are interesting/important.
>
> We merged Sections 4.4 and 4.5 into one section to make it less sparse.

---

### Review · Reviewer_P2Bz · 2024-11-01

**Summary Of Contributions:**

This paper presents DafnyBench, the largest benchmark designed for evaluating LLMs for formal software verification within Dafny. DafnyBench curates examples from open-source projects on GitHub, supplemented by two existing projects, to comprehensively assess the ability of various LLMs to generate Dafny annotations. Experimental results reveal that the most advanced LLMs achieve a success rate of approximately 68%. The study also highlights challenges in utilizing Dafny's feedback for model improvement and observes a significant performance decline with increasing program size or annotation complexity.

**Audience:**

Yes

**Claims And Evidence:**

Yes

**Requested Changes:**

- If feasible, consider extending the dataset size and incorporating additional evaluation tasks to broaden the benchmark’s applicability and rigor.
- Add a manual evaluation of the generated annotations to verify whether the LLM-generated annotations align with the ground truth in both correctness and detail.
- Table 2 reveals a significant disparity between the average and maximum values for various DafnyBench attributes. Displaying the distribution of these values would provide additional context and improve interoperability.
- Figure 2 is referenced in the related work section (page 2) but appears on page 6, making it challenging to follow. Moving it closer to its initial reference would enhance readability.
- Some statements require clarification. Many projects and benchmarks (particularly those using Coq and Isabelle theorem provers) focus on verifying software correctness (e.g., CompCert, SEL4), rather than purely mathematical theorem proving. Thus, statements like "there are only 66 + 153 = 219 for proving program correctness" are inaccurate and may overlook relevant related work.

**Strengths And Weaknesses:**

Strength:

- The paper is well-written and easy to follow.
- DafnyBench provides a broad collection of well-annotated Dafny programs, which can be a valuable resource for the formal verification community.

Weaknesses:

The authors provide a fair assessment of the paper's limitations, particularly regarding the benchmark’s limited size, potential data contamination, and the lack of evaluation on translating natural language into formal specifications. Specifically:

- The benchmark size (782 instances) remains very small. Expanding DafnyBench by manually constructing instances from other general programming benchmarks could further enhance its comprehensiveness.

- The paper focuses on a single annotation-filling task. Introducing additional evaluation tasks, especially for generating formal specifications directly from natural language, would significantly enhance the benchmark's utility for Dafny and formal verification research.

- The evaluation metric could be improved. Some annotations generated by LLMs may be simpler or more trivial than human-written ones, yet still technically correct. A more nuanced metric could better capture the quality and complexity of LLM-generated annotations.

---

> ### Author Response · Authors · 2024-11-18
> **Response to Reviewer P2Bz**
>
> We’re glad that you think DafnyBench can be a valuable resource for the formal verification community! We go through your suggestions in order.
>
> > The benchmark size (782 instances) remains very small. Expanding DafnyBench by manually constructing instances from other general programming benchmarks could further enhance its comprehensiveness.
>
> We agree that the benchmark size (782 programs) still remains very small. However, manually constructing instances might not help us expand the dataset by much. To add manually constructed instances requires someone to first formally verify the instances and have the ground truth programs — as part of our benchmark analysis (that we didn’t include in the paper), we asked one of our authors to formally verify some Dafny programs, some of which required the author on the order of 5 hours to verify even with LLM assistance. Since manually constructing an instance would be as time-consuming, we believe this wouldn’t help expand the dataset by much. Furthermore, both _Clover_ and _Dafny-Synthesis_ (the previously largest such datasets) contain manually constructed programs, but they have 66 and 153 programs respectively, which further shows the limited dataset expansion achievable with manual construction. However, future work might be able to augment / transform the programs in DafnyBench and/or ask a fine-tuned LLM to generate synthetic data, so we consider adding the following to Section 5.2:
>
> "Future work can augment the programs in DafnyBench by doing program transformations (e.g., by changing variable names in the programs, adding spurious code blocks, and changing order of lines that do not depend on each other), or expand the dataset by asking LLMs that are good at Dafny or fine-tuned on datasets like DafnyBench to generate synthetic data. However, this is beyond the scope of this paper."
>
> > The paper focuses on a single annotation-filling task. Introducing additional evaluation tasks would significantly enhance the benchmark's utility for Dafny and formal verification research.
>
> Adding additional tasks, especially generating formal specifications from natural language, would be very valuable for making the benchmark more useful. We chose fill_annotations as the task we focus on as we see it as a first step in developing more general formal verification benchmarks. In Section 5.4 and Appendix C, we discuss the potential of building bigger, more general benchmarks that can help with auto-verifying program synthesis, but actually incorporating more general tasks is beyond the scope of this paper.
>
> > The evaluation metric could be improved. Some annotations generated by LLMs may be simpler or more trivial than human-written ones, yet still technically correct. / Add a manual evaluation of the generated annotations to verify whether the LLM-generated annotations align with the ground truth in both correctness and detail.
>
> We appreciate your point that we should add a manual evaluation of LLM-generated annotations! While manually going through all the LLMs-generated annotations for the 782 programs might be hard to achieve in short term, we would like to point out that as part of the effort, we had anti-cheating checks that prevented LLMs from cheating — this is briefly mentioned in “Evaluation Metric” in Section 3.2, with examples in Appendix F. At the same time, if the Dafny verifier successfully verifies a file with LLM-filled annotations, at least we know that the LLM generated a coherent set of conditions that are true. We admit that this is not the same as having manual evaluation that can confirm the LLM-generated conditions are indeed desirable conditions, but being successfully verified increases the likelihood that they are desirable conditions.
>
> > Table 2 reveals a significant disparity between the average and maximum values for various DafnyBench attributes.
>
> Thank you for pointing this out! We replaced the table with histograms for methods / functions / lemmas / characters / annotation characters to better reflect the distributions of their quantities.
>
> > Figure 2 is referenced in page 2 but appears on page 6.
>
> Thanks for pointing out this inconvenience! We moved Figure 2 to page 3, so it’s now closer to where it’s first referenced (page 2).
>
> > Many projects and benchmarks (particularly those using Coq and Isabelle theorem provers) focus on verifying software correctness (e.g., CompCert, SEL4), rather than purely mathematical theorem proving.
>
> We focus on the data for formal verification, instead of any type of project in formal verification (e.g., CompCert is a verified compiler and doesn’t count in our mind as a proper machine learning benchmark). But we agree that saying “there are only 66 + 153 = 219 for proving program correctness” may overlook relevant work. Thus, we change it to “there is far less training data for formal verification; for example, there are only 66 + 153 = 219 for proving program correctness in the Dafny language.”

---

### Review · Reviewer_CmfJ · 2024-11-03

**Summary Of Contributions:**

This paper introduces DafnyBench, a new larger evaluation benchmark for formal software verification, motivated by similar large-scale benchmarks in mathematical theorem proving and program synthesis. The benchmark contains over 750 Dafny programs, expanding upon existing verification benchmarks (Clover and Dafny-Synthesis) which had only 66 and 153 programs, respectively.
They evaluate modern LLMs like GPT-4 and Claude 3 on DafnyBench in generating correct annotations, showing that the best model achieves a 68% success rate.

**Audience:**

Yes

**Claims And Evidence:**

Yes

**Requested Changes:**

included in the weaknesses above.

**Strengths And Weaknesses:**

**Strengths:**
* Addresses a clear gap in formal verification benchmarks compared to mathematical theorem proving and program synthesis domains.
* Comprehensive evaluation experiments on the proposed benchmark dataset with detailed analysis of different failure modes.
* Thorough documentation of benchmark construction and evaluation procedures.


**Weaknesses:**
* It's clear that the new benchmark dataset is a better dataset for the evaluation with respect to this task as it has larger size and more difficult problems. However, since majority of DafnyBench problems are scraped from GitHub repositories, there's a significant risk of data contamination - the tested LLMs (Claude 3, GPT-4) may have already been trained on these Dafny problems. This is particularly concerning compared to other benchmarks such as Dafny-Synthesis. There's a higher chance that LLMs have memorized these problems which are directly in Dafny Language in Github compared to the ones that have been translated from MBPP Python problems to Dafny in the Dafny-Synthesis benchmark (Misu et al., 2024).

* Beyond comparing dataset size and difficulty, it would be useful to see how LLMs perform across different data sources within DafnyBench (GitHub, Clover, Dafny-Synthesis). This could help to better analyze data contamination in LLMs' performance on GitHub-scraped problems versus the other benchmark sources.

* Limited discussion of mitigation strategies for data contamination


NOTE: Given that the paper's primary contribution is a benchmark dataset for evaluating LLM capabilities in formal software verification, I think that DMLR might be a better venue than TMLR.

---

> ### Author Response · Authors · 2024-11-18
> **Response to Reviewer CmfJ**
>
> We’re delighted to know that you think DafnyBench addresses a clear gap in formal verification benchmarks. And thanks for suggesting that DMLR might be a more suitable venue! Below, we go through each of your comments / suggestions.
>
> > Since majority of DafnyBench problems are scraped from GitHub repositories, there's a significant risk of data contamination - the tested LLMs (Claude 3, GPT-4) may have already been trained on these Dafny problems.
>
> We absolutely agree that scraping from GitHub introduces the concern of data contamination, as we pointed out in Section 5.2. Completely relieving the data contamination concern might be beyond the scope of this paper, but we added a further discussion of mitigation strategies in Section 5.2, as shown below in our response to the third point.
>
> > Beyond comparing dataset size and difficulty, it would be useful to see how LLMs perform across different data sources within DafnyBench (GitHub, Clover, Dafny-Synthesis). This could help to better analyze data contamination in LLMs' performance on GitHub-scraped problems versus the other benchmark sources.
>
> Thank you for suggesting that we can also compare performance across the different subsets (GitHub, _Clover_, _Dafny-Synthesis_)! Indeed, we did consider adding this analysis. However, _Clover_ and _Dafny-Synthesis_ both contain programs that have 1 - 2 short methods, thus on average are much shorter than those programs from GitHub. Therefore, we thought it would be a bit misleading to just compare the different subsets, and instead kept the comparison across different program lengths and annotation quantities. We hope this addresses your concern!
>
> > Limited discussion of mitigation strategies for data contamination
>
> This is a great point! While we briefly pointed out in Section 5.2 that data contamination is a concern for DafnyBench, we agree that we should discuss this concern in more detail, especially the mitigation strategies. We consider adding the following to the first paragraph of Section 5.2:
>
> "Future work can augment the programs in DafnyBench by doing program transformations (e.g., by changing variable names in the programs, adding spurious code blocks, and changing order of lines that do not depend on each other), or expand the dataset by asking LLMs that are good at Dafny or fine-tuned on datasets like DafnyBench to generate synthetic data. However, this is beyond the scope of this paper."

---

### Author Response · Authors · 2024-11-18
**Revised manuscript**

We want to thank all of the reviewers for very helpful feedback / suggestions! We uploaded a revised manuscript in response to the reviewer feedback.

---

### Decision · Action_Editor_SLUp · 2024-12-06

**Recommendation:** Accept as is

**Comment:**

The paper's main contribution is a new benchmark to test LLMs capabilities to annotate programs such that they can be formally verified (via the Dafny verifier in this case). Typically, providing such annotations is a tedious task for developers, as it means translating informal specifications into formal specifications - the potential speed-up through assistance and automation via LLMs is large. The paper adds a benchmark with over 750 programs, which improves over the largest previously published benchmarks by a large margin, and thus enables larger scale research. The paper also provides promising first results on the benchmark with GPT-4 and Claude 3 (under different prompting schemes), that show the potential for assistance and partial automation and also highlight the challenges for full automation with current state-ofthe-art models.

Overall, I think the paper makes an important contribution to drive forward and enable larger-scale research on automating formal verification with LLMs (verifiers exist, but the annotations are the bottleneck). While the size of the benchmark is still moderate, the authors put significant effort into collecting the dataset, which, together with the improvements in size compared to previously available benchmarks, is an non-trivial and significant contribution. The unavailability of licences for some of the examples (though all examples are publicly available) is a bit annoying, and may hinder adoption by commercial research labs, so if possible it would be great to try and rectify this in the future. Nonetheless, the work is clearly ready for publication and I have no open concerns or remaining issues from the reviewer discussion.

**Audience:**

All reviewers and I agree that the paper is interesting for a part of the TMLR audience.

**Claims And Evidence:**

Even in their initial reviews, all three reviewers agree that the paper makes concrete falsifiable claims that are supported by convincing and clear evidence. This still holds after the rebuttal and author discussion, and I agree with that assessment.